# Highly Sensitive Strain Sensor by Utilizing a Tunable Air Reflector and the Vernier Effect

**DOI:** 10.3390/s22197557

**Published:** 2022-10-06

**Authors:** Farhan Mumtaz, Muhammad Roman, Bohong Zhang, Lashari Ghulam Abbas, Muhammad Aqueel Ashraf, Yutang Dai, Jie Huang

**Affiliations:** 1Department of Electrical and Computer Engineering, Missouri University of Science and Technology, Rolla, MO 65409-0040, USA; 2Electrical Engineering Departmesnt, Sukkur IBA University, Sukkur Sindh 65200, Pakistan; 3Communications Laboratory, Department of Electronics, Quaid-i-Azam University, Islamabad 45320, Pakistan; 4National Engineering Laboratory for Fiber Optic Sensing Technology, Wuhan University of Technology, Wuhan 430070, China

**Keywords:** Vernier effect, strain sensor, hollow core fiber, Fabry–Perot interferometers

## Abstract

A highly sensitive strain sensor based on tunable cascaded Fabry–Perot interferometers (FPIs) is proposed and experimentally demonstrated. Cascaded FPIs consist of a sensing FPI and a reference FPI, which effectively generate the Vernier effect (VE). The sensing FPI comprises a hollow core fiber (HCF) segment sandwiched between single-mode fibers (SMFs), and the reference FPI consists of a tunable air reflector, which is constituted by a computer-programable fiber holding block to adjust the desired cavity length. The simulation results predict the dispersion characteristics of modes carried by HCF. The sensor’s parameters are designed to correspond to a narrow bandwidth range, i.e., 1530 nm to 1610 nm. The experimental results demonstrate that the proposed sensor exhibits optimum strain sensitivity of 23.9 pm/με, 17.54 pm/με, and 14.11 pm/με cascaded with the reference FPI of 375 μm, 365 μm, and 355 μm in cavity length, which is 13.73, 10.08, and 8.10 times higher than the single sensing FPI with a strain sensitivity of 1.74 pm/με, respectively. The strain sensitivity of the sensor can be further enhanced by extending the source bandwidth. The proposed sensor exhibits ultra-low temperature sensitivity of 0.49 pm/°C for a temperature range of 25 °C to 135 °C, providing good isolation for eliminating temperature–strain cross-talk. The sensor is robust, cost-effective, easy to manufacture, repeatable, and shows a highly linear and stable response for strain sensing. Based on the sensor’s performance, it may be a good candidate for high-resolution strain sensing.

## 1. Introduction

A fiber-optic strain sensor is an optical device that uses optical fiber technology to determine the strain on an object [1]. The strain experienced by the optical fiber is measured by measuring changes in the properties of light, such as intensity, wavelength, and state of polarization. These miniaturized sensors are based on optical fiber, which is often close to the diameter of human hair [2]. Due to their small size, fiber-optic sensors find practical applications in compact structures and tight spaces. Fiber-optic sensors have many advantages over electrical and electronic sensors, such as compact size, light weight, immunity to electromagnetic interference, low cost, stability, and high sensitivity [3]. Due to their versatile advantages, they have been made a part of many practical applications for various industrial sectors, such as defense, steel industry, structural health monitoring, civil engineering, bio-medical, aerospace, and marine engineering [4].

Fiber-optic strain sensors are reported in different designs and methodologies, such as fiber Bragg grating (FBG) [5,6], long period grating (LPG) [7,8], Mach–Zehnder Interferometer (MZI) [9,10,11], Michelson Interferometer (MI) [12,13,14], and FPI [15,16]. The FPI in fiber-optic strain sensors is most prominent, which can be realized by creating an air-gap/cavity to form two in-line reflective mirrors in an optical fiber. Sirkis et al. [17] initially reported a simple and compact extrinsic FPI with an ultra-high dynamic strain resolution of ~22 nε by employing single-wavelength time domain analysis. They also presented another airgap cavity-based in-line fiber etalon strain sensor [18], tested different lengths of FPI cavity ranging from 20 µm to 500 µm, and presented a slightly higher dynamic strain resolution of 30 nε through time domain analysis. The single-wavelength time domain analysis is typically based on detecting the intensity change at a specific wavelength and is limited to small-scale dynamic strain sensing due to the nonlinear response and limited free spectral range. Cibula et al. [18] reported a strain sensor for quasi-distributed measurement using optical time domain refractometry technology. They measured transmission loss of less than 1 dB with an ability to measure strain up to ±2500 μm/m, which makes the sensor suitable for quasi-distributed sensing when 10 in-line FPIs were deployed in series. The fabrication method is challenging due to the inconsistent process that involves chemical etching and fusion splicing of the core region of the optical fiber to assemble FPI. Ran et al. [19] reported a strain sensor based on a miniature in-line photonic crystal fiber (PCF) etalon air-cavity. The air cavity length was 45.6 μm, fabricated by 157 nm laser micro-machining. The reported sensor presented a strain sensitivity of 0.33 nm/µε, with a measurement range limited to 800 µε. The limited measurement range is due to the collapse of the nanoholes in a PCF after stretching. Pevec et al. [20] reported a strain sensor by introducing a long active air cavity with an arbitrary-length air cavity and attained a strain-resolution of 1 µε in the range of 0 to 3000 µε. The sensor design is unique but needs a special optical fiber and a complex etching procedure, making the fabrication time-consuming. Tian et al. [21] reported FPI-based strain sensors by utilizing a concave core PCF. They fabricated four sensors with different lengths of FPIs, and the FPI length of 4.85 µm presented maximum strain sensitivity of 31.58 pm/µε in the range of 0 to 1600 µε. They used a different arc power to deform the end facet of concave-shaped PCF at one side; therefore, it can be sandwiched between SMFs to realize the airgap-based FPI. Deng et al. [22] reported an all-in-line fiber strain sensor by utilizing PCF and assembling a micro-bubble-based air cavity. They obtained a maximum strain sensitivity of 2.3 pm/µε in the range of 0 to 1850 µε at 800 °C. Zhao et al. [23] reported a strain sensor using a short segment of etched mirco-notch graded index multi-mode fiber spliced with a lead-in SMF to realize an air-cavity-based FPI. The sensor presented a strain sensitivity of 7.82 pm/µε and temperature sensitivity of 5.01 pm/°C in the strain measurement range of 0 to 845 µε and temperature measurement range of 0 to 90 °C, with a large temperature strain cross-talk and strain measurement error of 19 µε within a 30 °C temperature change. Tian et al. [24] reported a temperature and strain sensor that comprises a 246 µε-long HCF silica tube sandwiched between SMFs. The sensitivity of the sensor is low, but a detailed annealing procedure was successfully demonstrated for examining the HCF stability at higher temperatures, i.e., 1000 °C, under different strains, i.e., from 0 to 1000 µε. Zhang et al. [25] reported a strain sensor based on a FBG cascaded with FPI. The FPI cavity is tuned by inserting a tapered SMF end facet into the HCF and an FBG is inscribed into the tapered region of SMF to compensate for the temperature response. The sensitivity of the sensor is low (i.e., 2.1 pm/µε), but the sensor is capable to compensate for temperature–strain cross-talk. Tian et al. [26] explored a strain sensor with low temperature–strain cross-talk by embedding a micro-sphere into the tapered HCF. The paper demonstrated a flexible cavity length of FPI by different waists of tapered HCF. The sensor presented a maximum strain sensitivity of 16.2 pm/µε, with a strain resolution of 1.3 µε with limited strain measurements (i.e., 0 to 160 µε) due to the fragility of the tapered HCF.

In contrast to a single FPI sensor, the Vernier-effect-based cascaded or parallel FPI sensors have been demonstrated to further enhance the sensor’s measurement sensitivity and resolution [27,28,29,30,31,32,33]. By exploiting the small-scale difference between the ruler and Vernier scale, the VE was originally used for high-precision length measurements. Similarly, VE can be employed to obtain ultra-high sensitive sensing structures. The optical VE is usually generated by superimposing interferograms of a reference interferometer and a sensing interferometer. Unlike conventional interferometry where the spectrum of the sensing interferometer is monitored, VE-based sensors rely on tracking shifts in the envelope of the superimposed spectrum. VE-based fiber-optic sensors are prominent in optical sensing due to their highly sensitive response to different physical quantities. VE-based FPIs are reported to attain maximum sensitivity of a fiber-optic sensor for various applications, such as monitoring strain, temperature, refractive index, humidity, and pressure. Cleas et al. [27] reported two cascaded ring resonators that work analogously to a Vernier-scale and obtained relatively high sensitivity for refractive index measurement. The authors described the concept of VE-based sensors and presented a very detailed analysis. Liu et al. [28] explored a high-sensitivity strain sensor based on in-fiber improved FPIs by fabricating cascaded bubbles to generate VE and attained a strain sensitivity of 6.6 pm/με. Although their proposed sensor obtained high sensitivity, the sensor’s repeatability is quite challenging due to the non-uniformity of bubbles-based air cavities. Abbas [29] reported a VE-based strain sensor and obtained optimum strain sensitivity of 37.3 pm/με, but the sensor showed a limited detection range of axial strain, which was caused by the tapered structure. Nan et al. [30] reported a VE-based strain sensor with a configuration of closed air cavity with open air cavity. The proposed sensor obtained maximum strain sensitivity of 43.2 pm/με in the range of 0 to 1750 με, which is a substantial improvement in the sensor’s performance in terms of sensitivity, but lateral off-setting of single-mode fiber (SMF) in sensing FPI increases the risk of fragility. Andre et al. [31] explored an optical harmonic VE-based strain sensor and used a first, second, and third harmonic algorithm to obtain ultra-high sensitivity values of 27.6 pm/με, 93.4 pm/με, and 59.6 pm/με, respectively. However. the sensor requires extra broad bandwidth to reach ultra-high sensitivity, i.e., 1200 nm to 1600 nm. Deng et al. reported a VE-based strain sensor by creating in-fiber reflection mirrors using femto-second laser micromachining and obtained strain sensitivity of 28.11 pm/με in the range of 0 to 1400 με, but such sensors require expensive equipment for fabrication. Wu et al. [32] reported a strain sensor design using demodulation techniques of coarse wavelength and generated VE. The sensor reached a strain sensitivity of 18.36 pm/με, which is substantially higher, but coarse wavelength demodulation increases the margin of error for critical applications. Zang et al. [33] reported a highly sensitive fiber-optic sensor based on two cascaded intrinsic fiber FPIs for strain and magnetic field measurements. The sensor reached a strain sensitivity of 47.14 pm/but showed a demerit of elongated sensing FPI and reference FPI, i.e., 9.724 mm and 9.389 mm; thus, it cannot be utilized for miniaturized applications. To address the present need for industrial practical applications, still, there is a large market waiting for the development of fiber-optic sensors that have the capabilities of compactness, robustness, repeatability, stability, cost effectiveness, high sensitivity, and ease of fabrication.

In this paper, a novel tunable air-reflector-based strain sensor is proposed that generates an efficacious VE. The sensor has an advantageous feature of ultra-low cross-talk between strain and temperature. The isolation of the reference FPI of the sensor provides an admirable and stable response. The sensor structure is super robust and capable of measuring an extensive range of axial strain. Beam propagation simulation is performed to determine the excitation levels of induced modes in the interferogram. Commercially available off-the-shelf optical fibers are used to design and fabricate the proposed sensor, i.e., SMF and HCF. The sensing structure is miniaturized due to only using a few microns of the HCF segment. As the cavity length of the sensing FPI and reference FPI are accurately determined by using the computer-controlled Glass processor (GPX-3400) platform along with LDC-401 cleaver, the sensor is formed with precision. Further, highly optimized strain sensitivity is readily obtained by using a VE subject to the limitation of the source bandwidth (i.e., 80 nm). The strain sensitivity of the proposed sensor is obtained as 0~23.9 pm/με within the measurement range of 0 to 3000 με, which is approximately 24 times higher than FBG (~1.01 pm/με) [25].

The paper is organized as follows: Section 2 describes working principles, Section 3 demonstrates the fabrication of the sensor, Section 4 offers the simulated beam propagation profile of the proposed sensor, Section 5 illustrates the experimental results and discussion, and, finally, Section 6 provides a conclusion.

## 2. Working Principle

The schematic of the sensing and reference FPIs is shown in Figure 1a,b, respectively. To achieve the VE, the two FPIs are cascaded and the light is injected through a lead-in SMF. There are four mirrors in the proposed sensing structure, and, when the light reaches the end facet of HCF (fourth mirror), the majority of light is reflected back and becomes a part of the VE interferogram, which can be expressed as:(1)$$I=I_{1}+I_{2}+I_{3}+I_{4}+2\sqrt{I_{1}I_{2}}\cos\left(\phi_{r}\right)+2\sqrt{I_{3}I_{4}}\cos\left(\phi_{s}\right)$$
where *I*_1_, *I*_2_, *I*_3_, and *I*_4_ are the reflected light intensities from four different mirrors M_1_, M_2_, M_3_, and M_4_, respectively. *ϕ_r_* and *ϕ_s_* are the cosine angles measuring phase differences of the reference and sensing FPIs, and are estimated as:(2)$$\phi_{r}=\frac{4\pi n_{r}L_{r}}{\lambda }$$
(3)$$\phi_{s}=\frac{4\pi n_{s}L_{s}}{\lambda }$$
where *λ* is the operating wavelength. In order to generate efficacious VE, two optical lengths are formed that are similar to each other but not equal. Thus, the cavity length of the reference and sensing FPIs can be determined as:(4)$$L_{r}=\frac{\lambda_{m}\lambda_{m+1}}{2n_{r}(\lambda_{m}-\lambda_{m+1})}$$
(5)$$L_{s}=\frac{\lambda_{k}\lambda_{k+1}}{2n_{s}(\lambda_{k}-\lambda_{k+1})}$$
where *n_r_*, *n_s_* are refractive indices of reference and sensing FPI cavities. *λ_m_* and *λ_m_*_+1_, *λ_k_* and *λ_k_*_+1_ are the adjacent resonant dips in the wavelength interferogram of the corresponding reference and sensing FPIs, respectively, and, the wavelength of the resonant of the corresponding FPIs can be calculated as:(6)$$\lambda_{m}=\frac{4\pi n_{r}L_{r}}{2m+1},\;m=0,1,2,\ldots \ldots \mathrm{for}\;\mathrm{reference}\;\mathrm{FPI}$$
(7)$$\lambda_{k}=\frac{4\pi }{2k+1}n_{s}L_{s},\;k=0,1,2,\ldots \ldots \mathrm{for}\;\mathrm{sensing}\;\mathrm{FPI}$$
where *L_r_* and *L_s_* are the cavity lengths of the reference and sensing FPIs, and their medium is air, which assumes the refractive index: *n_r_* = *n_s_* = 1. The reference FPI is isolated, meaning only the sensing FPI will experience axial strain resulting in drifts in wavelength that can be expressed as:(8)$$\delta \lambda_{k}=\frac{4\pi \Delta L_{s}}{2k+1}=\frac{\lambda_{k}}{n_{s}L_{s}}$$
where Δ*L_s_* is the change in sensing FPI cavity length due to applied strain. Above Equation (8) shows that the strain sensitivity corresponds to the sensing FPI, and changes in cavity length due to applied strain are larger than a single FPI can sense. The free spectral range (*FSR*) of the sensing and reference FPIs can be estimated as:(9)$$FSR_{r}=\lambda_{r}(m-1)-\lambda_{r}(m)=\frac{\lambda_{r}^{2}}{2n_{r}L_{r}}$$
(10)$$FSR_{s}=\lambda_{s}(k-1)-\lambda_{k}(k)=\frac{\lambda_{s}^{2}}{2n_{s}L_{s}}$$
where *λ_r_* and *λ_s_* are the resonant wavelength dips of reference and sensing interferograms, respectively. According to VE theory, the envelope *FSR_e_* can be estimated as:(11)$$FSR_{e}=\left|\frac{FSR_{s}\cdot FSR_{r}}{FSR_{s}-FSR_{r}}\right|$$

By employing VE, the sensor’s sensitivity can be substantially improved by tracking the drift of the envelope depression. Thus, an amplification factor *M* of the VE-satisfying sensor can be estimated as:(12)$$M=\frac{FSR_{s}}{FSR_{r}-FSR_{s}}$$
and drift in wavelength of the envelope can be expressed as:(13)$$\delta \lambda_{e}=M\times \delta \lambda_{k}=\frac{FSR_{s}}{FSR_{r}-FSR_{s}}\times \frac{\lambda_{k}}{L_{s}}\Delta L_{s}$$

From Equation (13), it can be inferred that VE phenomena can substantially enhance the sensitivity of the sensor by M times. Thus, the proposed sensor will be able to measure axial strain with high sensitivity by employing VE in the sensing structure.

## 3. Fabrication of the Sensor

The proposed sensor is designed with two different FPIs. The reference FPI consists of HCF, and the sensing FPI consists of an HCF sandwiched between SMFs. An SMF-28e with core and cladding diameter of 8.2 µm and 125 µm and refractive indices of 1.4682 and 1.4672, respectively, is used as a lead-in fiber. An HCF with inner and outer diameters of 55.1 µm and 125 µm, respectively, is used to fabricate the sensing FPI. The material of HCF is pure silica, and its refractive index is 1.444. The sensor is fabricated by a simple process of splicing and cleaving. The GPX-3400 and LDC-401 cleaver are used in the fabrication of the proposed sensor, as shown in Figure 2. The fabrication process is as follows: the sensing FPI is formed by splicing of SMF–HCF–SMF in the form of concatenation. The SMF is cleaved and then spliced with a piece of HCF, as shown in Figure 3a,b. Then, the HCF is cleaved at 400 μm with the LDC-401 cleaver, which has the ability to cleave at a micro-scale with high precision. After cleaving the HCF, it is spliced with SMF so that the SMF–HCF–SMF structure is formed. The GPX-3400 is a computer-aided device that provides high precision while splicing optical fibers, and the splicing is performed with controlled filament burning. The SMF and HCF are spliced with filament power of 0~70 W, a pre-gap distance of 8.0 μm, a pre-push distance of 5.0 μm, a hot push distance of 14.0 μm, and a filament burning duration of 5 s. The reason for using low power while splicing the HCF with SMF is that HCF has a hollow air core, and, if high power is used for splicing, the HCF collapses and subsequently deforms the fiber shape. The reference FPI is formed by placing a cleaved SMF fiber on the GPX-3400 fiber holding blocks. The fiber holding blocks of the GPX-3400 platform are computer-controlled, which helps to align the fibers. The reference FPI is a tunable air reflector that can be adjusted. To examine the sensor’s performance corresponding to VE, three different samples of air reflectors are tuned to form reference FPIs, i.e., S-1 = 375 μm, S-2 = 365 μm, and S-3 = 355 μm. The schematic fabrication process of the proposed sensor is shown in Figure 3. The microscopic images of longitudinal cross-sections of tunable reference FPIs, the sensing FPI, and the transverse cross-sections of SMF and HCF are shown in Figure 4a–d, respectively. Microscopic images are taken by GPX-3400, which has a CCD camera connected to the computer, as shown in Figure 2.

## 4. BPM Profile of the Sensor

To further analyze the light beam propagation in the proposed sensing FPI, a three-dimensional beam propagation module (BPM) is used by Rsoft software. Figure 5a demonstrates the slice view of beam propagation in the *xz*-plane for the sensing FPI, where yellow and grey colors depict the SMF and HCF, and their parameters are consistent with the original datasheets of the optical fibers. Figure 5b describes the modal profile of the beam in the *xz*-plane, which is propagated through SMF–HCF–SMF. The inset of Figure 5b depicts the *xy*-mode profile propagating through the deployed optical fibers corresponding to the z-direction. Cross-section “A” is taken at z = 10 µm, specifying the fundamental mode profile with an effective mode index of 1.46326, which is injected via lead-in SMF. Cross-section “B” is taken at z = 300 µm, which shows the E_x_ mode profile with an effective mode index of 1.465242 at the middle of HCF, and it can be observed that light started spreading via the hollow core to the cladding region. Cross-section “C” is taken at z = 500 µm, where the beam of light exits the HCF segment. The light is further dispersed into the cladding region and obtains a higher order (HE) mode with an effective mode index of 1.466397. Finally, cross-section “D” is taken at z = 900 µm with an effective mode index of 1.46675. It can be seen that the light intensity further deteriorates while exiting from the HCF due to the air reflector. At this point, HE modes are more prominent. Figure 5c displays the light intensity profile of the light beam, which is injected through the lead-in SMF. The profile provides a better understanding of the intensity of light distribution as it propagates through the sensing structure. Consequently, a significant portion of the light is reflected from the cross-section “C” region and becomes part of the interferogram. Figure 5d provides a three-dimensional view of the modal profile for the proposed sensing structure, which further classifies the mode dispersions while the light beam is exiting from HCF and eventually excites HE modes.

## 5. Experimental Results and Discussion

A schematic of the experimental setup for the axial strain measurement is shown in Figure 6. A broadband source (Thorlabs, Model # ASE-FL7002-C4) with a narrow bandwidth of 1530 nm to 1610 nm, optical spectrum analyzer (Model # AQ6317B), a laptop for the acquisition of data, and a 3dB coupler are used to obtain the reflection spectrum of the proposed sensor. For reference, axial strain measurements were taken with the sensing FPI structure without the cascading tunable reference FPI. The sensing FPI response to different values of axial strain from 0 to 3000 µε are recorded and plotted as wavelength shift versus applied axial strain, as shown in Figure 7. The FSR of the sensing FPI was obtained as 2.985 nm. In contrast, the strain sensitivity of a single sensing FPI is obtained as 1.74 pm/µε with excellent linear fit correlation. The inset of Figure 7 depicts the spectrum evolution of the sensing FPI dip, which shifts linearly with the increasing value of axial strain.

Thereafter, to realize the VE of the proposed sensor, the sensing FPI was cascaded with the tunable reference FPI. Three different samples of tunable reference FPIs were used to analyze the VE of the proposed sensor, i.e., S-1, S-2, and S-3. Figure 8 depicts the reflection interference spectrum of S-1, S-2, and S-3. The curve fitting method was used to draw the upper and lower envelopes over the VE interference pattern. The measured envelope *FSR_e_* values of S-1, S-2, and S-3 are 41.45 nm, 31.33 nm, and 24.96 nm, as shown in Figure 8a–c, which are closely equivalent to the theoretically calculated values of 41.74 nm, 30.97 nm, and 24.62 nm, respectively. The minimal error is caused by the measuring equipment, which is negligible.

The axial strain response of S-1 is measured by using the experimental setup as shown in Figure 6. The spectral evolution of S-1 is recorded by applying stress, which causes axial strain from 0 µε to 2600 µε, with an axial strain step of 200 µε. The envelope dip produces a red-shift (the change in wavelength toward a longer wavelength) in the spectra corresponding to increasing axial strain. The wavelength versus envelope data of S-1 is plotted in Figure 9. It can be seen that, by using VE, S-1 strain sensitivity is 23.9 pm/µε, which is almost 14 times higher than that of a single sensing FPI. The S-1 measurement also provides an excellent linear fit correlation function with R^2^ = 0.99751. The measured strain values match well with theoretical analysis, and a small error is observed, which is induced by equipment measurement error, as provided in Table 1. Further, in order to realize the amplification factor of the proposed sensor, different cavity lengths in the cascaded FPIs are tested, i.e., S-2 and S-3. Air cavities for the reference FPIs are formed using a tunable cavity platform, where the cavity lengths for S-2 and S-3 are 365 µm and 355 µm, respectively. The FSR of the individual reference FPIs for S-2 and S-3 is measured as 3.28 nm and 3.36 nm, respectively. The spectral evolution is presented in the inset of Figure 10. S-2 exhibits axial strain sensitivity of 17.54 pm/µε, with an excellent linear correlation function of R^2^ = 0.99905 in the strain measurement range of 0 µε to 3000 µε, and produces a red-shift with increasing axial strain, as shown in Figure 10. The strain sensitivity of S-2 is 10 times higher than that of the single sensing FPI. Finally, S-3 was tested to measure its response to axial strain. As with S-1 and S-2, S-3 also produced a red-shift in the spectral evolution with increasing axial strain. The plot of S-3-exhibited wavelength versus dip variation due to axial strain is shown in Figure 11. S-3 exhibits axial strain sensitivity of 14.11 pm/µε, which is 7.98 times higher than the single sensing FPI, and shows an excellent linear correlation function of R^2^ = 0.99901 in the strain measurement range of 0 µε to 3000 µε, as shown in Figure 11.

It can be inferred that, by reducing the cavity length of the reference FPIs of S-1, S-2, and S-3, the *FSR_e_* of each sample is reduced and exhibits a substantial decrease in strain sensitivity. S-1 reached maximum sensitivity of 23.9 pm/µε subjected to the limitation of source bandwidth, i.e., 80 nm. However, the strain sensitivity of the sensor can be further enhanced by increasing the bandwidth of the source. Nevertheless, the proposed sensor showed a highly sensitive response to strain. It provided measurements over a wide range of axial strain by employing a narrow-bandwidth lightsource, which is an advantageous feature of the sensor. The wide range of strain measurement confirms that it is a mechanically robust sensing structure. The length of the sensor’s structure is 800 microns, which further confirms its compactness. The obtained sensitivity and *FSRs* of the three samples are listed in Table 1.

The repeatability of the sensor is also tested after taking the strain measurements. Figure 12 exhibits the repeatability of the sensor. The up-strain measurements were taken in the strain measurement range of 0 µε to 3000 µε, which were then followed by down-strain measurements over the same range. It is observed from both up and down-strain measurements that the envelope dips of three samples are very stable and approximately return to the same position. However, a minimal fluctuation is observed in linearity, which is negligible and is caused by the strain measurement error of equipment. A comparison of the proposed sensor is made with earlier reported sensors, which are listed in Table 2. It can be ascertained from the comparison that effective use of VE in sensing architecture can significantly enhance the sensitivity of the sensor, and the proposed sensor demonstrated higher sensitivity with a wide strain detection range. However, Refs. [29,33] reported higher strain sensitivities but presented a limited detection range, i.e., less than 444 µε and 60 µε, respectively. Moreover, the proposed structure is simple and easy to fabricate compared to the reported structures, which required laser inscription [5,7] and difficult fabrication techniques [16,28] for realization. Additionally, the proposed sensor exhibits far superior sensitivity than the reported sensors in Refs. [6,8,10,11,14,28]. Furthermore, the proposed sensor presents a strain resolution of ~0.83 µε based on the demodulation resolution of the interferograms of ~0.02 nm, which is significantly improved compared to the conventional fiber-optic strain sensors, such as FBGs (17.54 µε) [6], LPGs (4.57 µε) [8], MZI (2.94 µε) [11], hybrid structure (1.72 µε) [14], and tapered FPI (1.3 µε) [26]. Moreover, the single FPI structures are reported [17,18] for high strain resolutions (i.e., up to 30 nε) using time domain analysis, and, typically, single-wavelength time domain analysis is limited to dynamic small strain sensing due to non-linear response and limited FSR.

The temperature response of the proposed sensor was also tested. To test the temperature response, the sensing FPI is placed into a heating furnace whose temperature error is about 0.1 °C. The temperature measurements were recorded from 25 °C to 135 °C with a temperature step of 10 °C, as shown in Figure 13. The sensor exhibits ultra-low temperature sensitivity of 0.49 pm/°C with a good linear correlation function of R^2^ = 0.9982, as shown in Figure 14. The low sensitivity of the sensing FPI cannot be further amplified with VE because both have nearly identical cavities and would not influence a large change in refractive index when exposed to an analogous temperature environment. Therefore, ultra-low temperature sensitivity is a superior aspect of the proposed sensor, which helps to eliminate the temperature cross-talk with strain measurements. The proposed sensor exhibits temperature–strain cross-talk as low as 0.0205 με/°C. It shows that the proposed sensor is fully capable of compensating for temperature variations when it is under different strains (i.e., 0 µε to 3000 µε). Thus, the cascaded FPIs sensing structure with VE has many advantages for numerous practical applications.

## 6. Conclusions

In summary, a highly sensitive strain sensor based on tunable cascaded FPIs was experimentally demonstrated. Cascaded FPIs efficaciously generate the Vernier effect in the sensing structure. The novelty of the sensor is to tune a desired air reflector cavity for the reference FPI to obtain higher sensitivity and improve performance. The Vernier effect enhances the sensor’s sensitivity by 13.73 times compared to a single sensing FPI when the reference FPI length is 375 μm. The robustness of the sensor enables measurement of large strains. The sensor is flexibly designed for narrow-bandwidth lightsources and presents low cross-talk with temperature, as low as 0.0205 με/°C. The unique features of the proposed sensor are easy fabrication, cost-effectiveness, high sensitivity, mechanical robustness, repeatability, compactness, and precise control of the FPI’s cavity, which makes it suitable for numerous industrial applications.

## Figures and Tables

**Figure 1 sensors-22-07557-f001:**
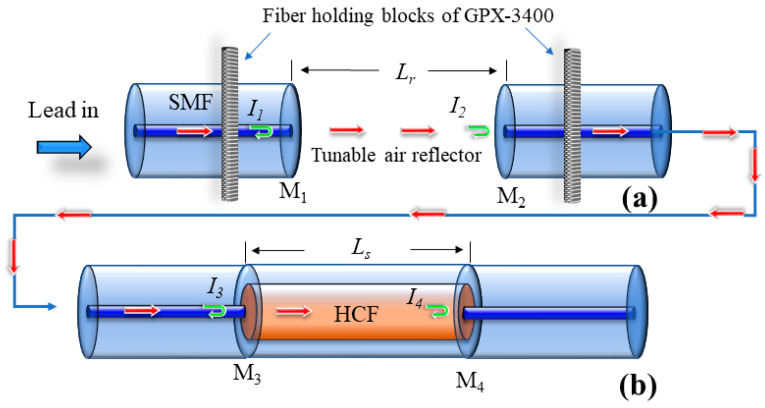
Proposed sensor: (**a**) reference FPI, (**b**) sensing FPI.

**Figure 2 sensors-22-07557-f002:**
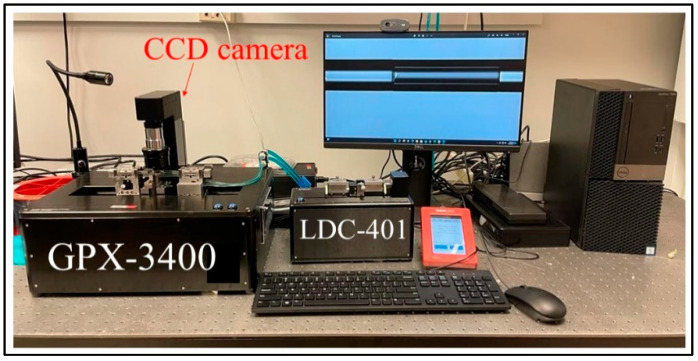
Splicing and cleaving setup for fabrication of the proposed sensor.

**Figure 3 sensors-22-07557-f003:**
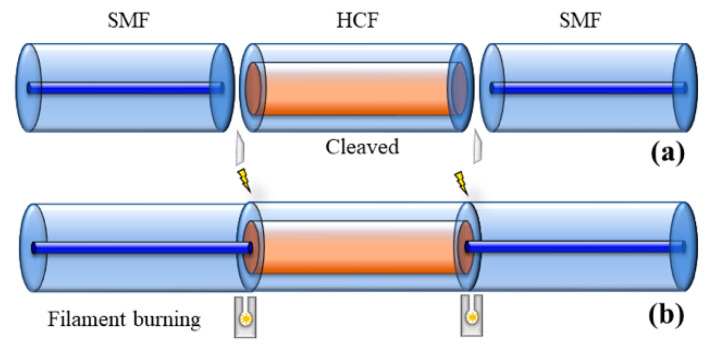
The fabrication process of sensing FPI (**a**) cleaved fibers, (**b**) spliced fibers.

**Figure 4 sensors-22-07557-f004:**
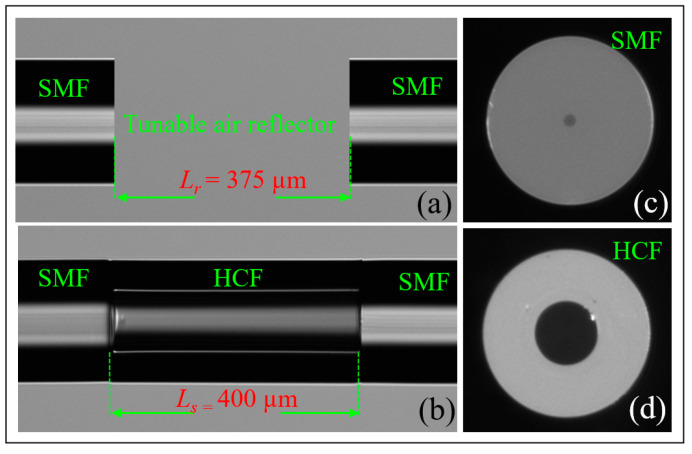
Microscopic images of the proposed sensor: (**a**) longitudinal cross-section of (**a**) tunable reference FPI (**b**) sensing FPI, and transverse cross-section of (**c**) SMF (**d**) HCF.

**Figure 5 sensors-22-07557-f005:**
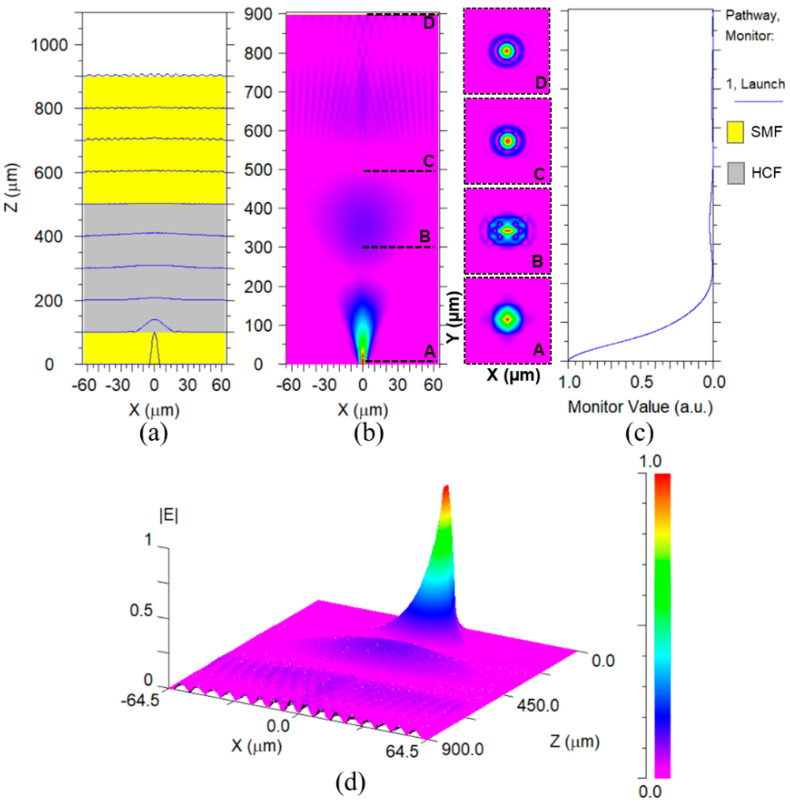
BPM profile of the proposed sensor: (**a**) slice view in the *xz*-plane, (**b**) modal profile in the *xz*-plane and insets show the *xy*-plane mode field at different z, (**c**) monitor pathway of light intensity propagating through the sensor, and (**d**) 3-D mode field distribution.

**Figure 6 sensors-22-07557-f006:**
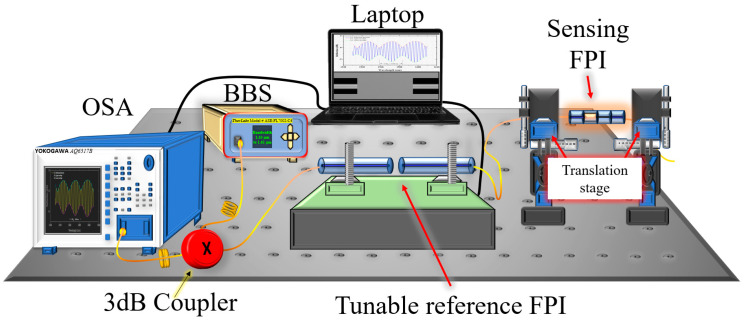
Schematic experimental setup for strain measurements.

**Figure 7 sensors-22-07557-f007:**
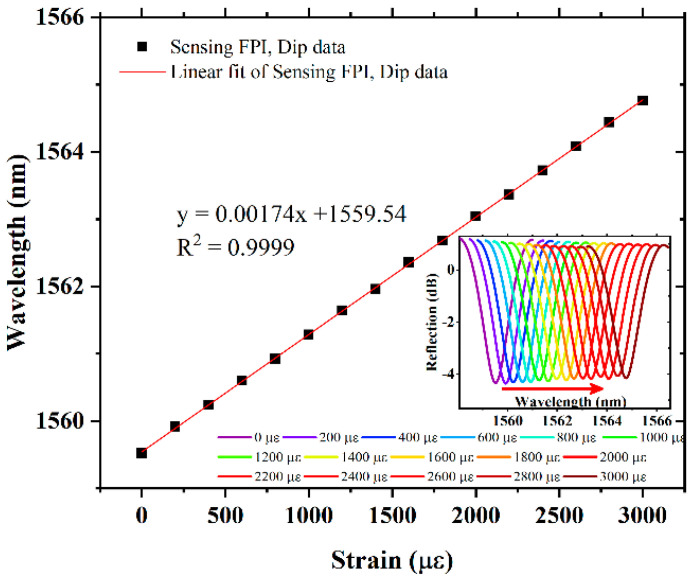
Axial strain measurement response of single sensing FPI.

**Figure 8 sensors-22-07557-f008:**
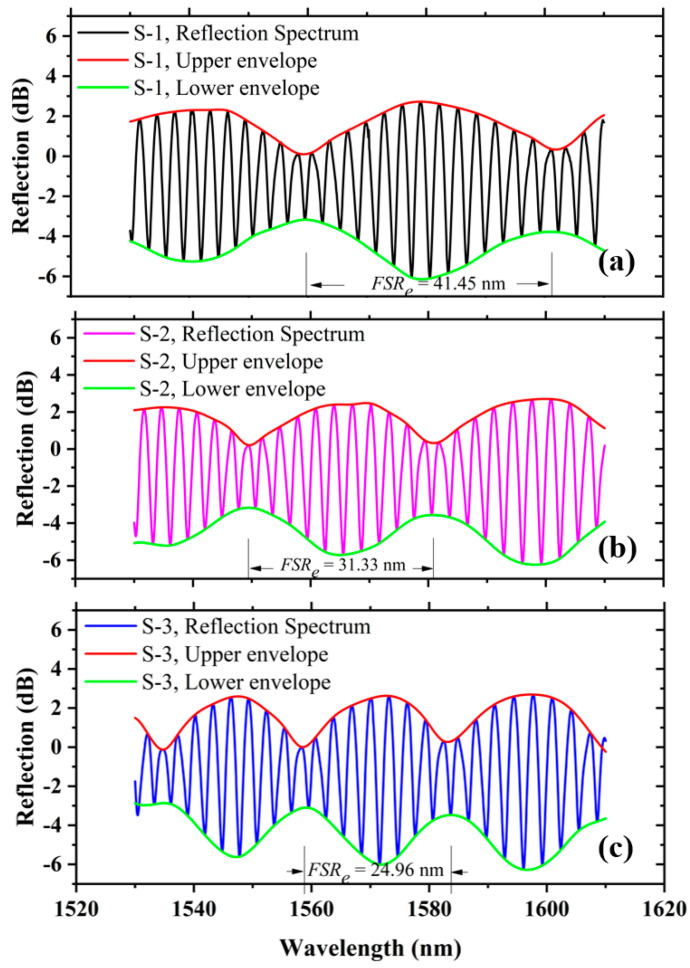
*FSR_e_* of cascaded FPIs (**a**) S-1, (**b**) S-2, (**c**) S-3.

**Figure 9 sensors-22-07557-f009:**
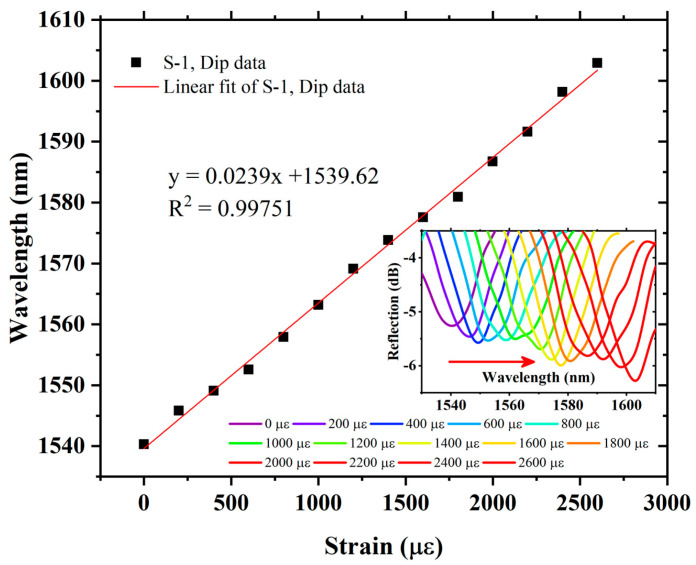
Sensitivity and linear correlation function of S-1, and inset exhibits spectral evolution of envelope dip data.

**Figure 10 sensors-22-07557-f010:**
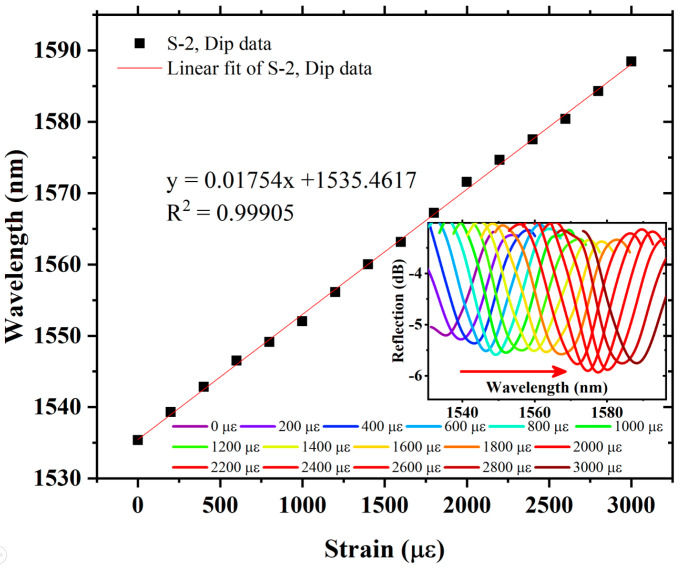
Sensitivity and linear correlation function of S-2, and inset exhibits spectral evolution of envelope dip data.

**Figure 11 sensors-22-07557-f011:**
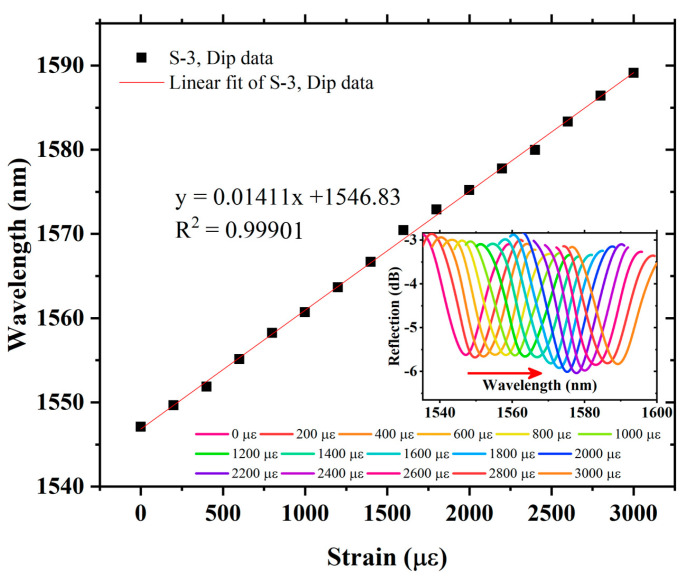
Sensitivity and linear correlation function of S-3, and inset exhibits spectral evolution of envelope dip data.

**Figure 12 sensors-22-07557-f012:**
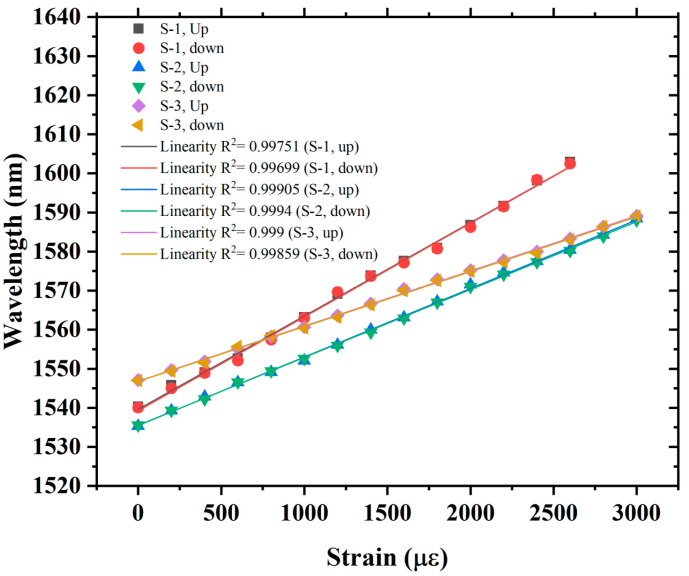
Repeatability test of the proposed sensor.

**Figure 13 sensors-22-07557-f013:**
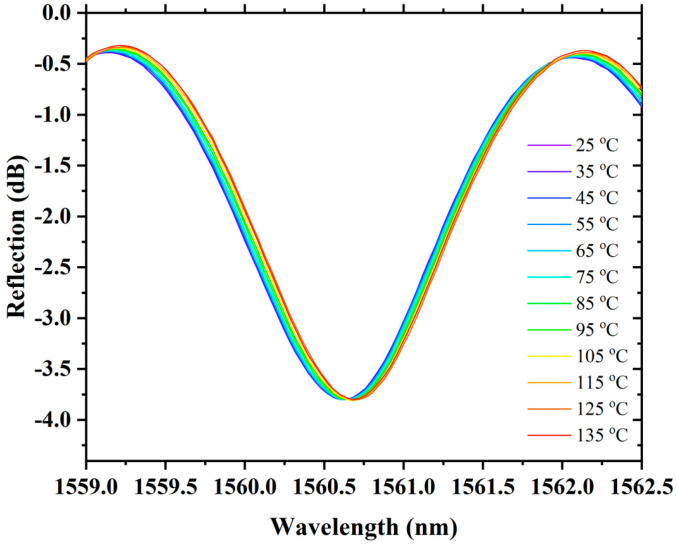
Spectral evolution of the proposed sensor with temperature rise.

**Figure 14 sensors-22-07557-f014:**
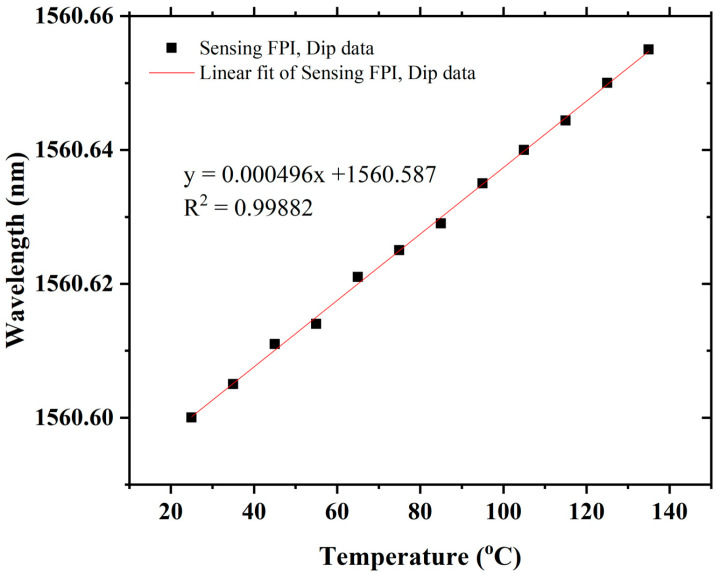
Sensitivity and linear correlation function.

**Table 1 sensors-22-07557-t001:** *FSR* and sensitivity of the proposed sensor.

Sample	*FSR_r_ * (nm)	*FSR_s_ * (nm)	Theoretical		Experimental		Sensitivity
			*FSR_e_* (nm)	Mag. Factor M	*FSR_e_* (nm)	Mag. Factor M	(pm/µε)
S-1	3.20	2.985	43.90	13.71	41.45	13.73	23.90
S-2	3.28	2.985	33.04	10.07	31.33	10.08	17.54
S-3	3.35	2.985	27.08	8.07	24.96	8.10	14.11

**Table 2 sensors-22-07557-t002:** Comparison of the proposed sensor with earlier reported sensors.

Reference	Sensing Structure	Strain Sensitivity (pm/µε)	Strain Measurement Range (µε)	Temperature Sensitivity (pm/°C)	Temperature Measurement Range (°C)
[6]	FBG	1.14	0–5000	11.2	25–200
[7]	PCF LPG	−7.6	0–800	3.91	20–90
[8]	LPG	4.37	0–700	85.0	0–60
[10]	MZI	1.19	0–800	137.6	20–90
[11]	MZI with Microcavity	6.8	0–700	-	-
[14]	Hybrid	11.6	0–295	11.6	30–210
[16]	Micro-FPI	2.39	0–9800	0.9	30–100
[28]	Bubble-based FPI	6.0	0–1000	1.1	100–600
[29]	Cascaded Tapered FPIs	37.3	0–444.4	0.7	30–80
[33]	Cascaded FPIs	47.14	0–60	-	-
This work	Tunable Cascaded FPIs	23.9	0–3000	0.49	25–135

## Data Availability

Not applicable.

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
