# Peer review of "Highly Sensitive Strain Sensor by Utilizing a Tunable Air Reflector and the Vernier Effect"

_sensors, 2022, doi:10.3390/s22197557_

Round 1
Reviewer 1 Report
In this paper, the authors presented a highly sensitive strain sensor based on tunable cascaded Fabry-Perot interferometers (FPI), the novelty of which lies in the tuning of the required air reflector cavity for the reference FPI to obtain higher sensitivity and improved performance. Cascaded FPIs comprise a sensing FPI and a reference FPI that effectively produce a Vernier Effect (VE). Unlike conventional interferometry, which monitors the spectrum of a sensing interferometer, VE-based sensors rely on tracking changes in the superimposed spectral envelope, and in optical sensing, VE-based fiber optic sensors are notable for their highly sensitive response to different physical quantities. This paper is somewhat novel and provides a specific study of the subject, but there are still some problems with this paper. The following are the issues in this manuscript.
(1) Lines 20-22 and 328-329, the text mentioned that the proposed sensor exhibits an optimum strain sensitivity of 23.9 pm/με over the range 0 με to 3000 με , which is 13.73 times higher than the strain sensitivity of 1.74 pm/με for a single sensing FPI. As can be seen below, by using VE, the S-1 strain sensitivity is 23.9 pm/με , but the strain range for S-1 is 0 με to 2600 με . Not only this, but S-2 and S-3 are not 13.73 times more sensitive than the strain sensitivity of a single sensing FPI. It is recommended that it should be made clear that it is S-1 (375μm) that exhibits the best sensitivity of 23.9 pm/ με in its range, and not all air reflector samples achieve the best strain sensitivity of 23.9 pm/ με .
(2) Lines 99-100 and 168, the strain sensitivity range is suggested to be modified to 0 ~ 23.9 pm/ με. It was also mentioned in the text that the strain sensitivity of the proposed sensor is 24 times higher than that of the FBG, and it is suggested that the value of the FBG sensitivity be given. the filament power of the SMF and HCF after splicing is suggested to be modified to 0 ~ 70 W.
(3) Eqs. (2), (3), did not explain the parameter λ; Eqs. (9), (10), did not explain the parameters λr and λs; Eq. (11), the denominator was not fully displayed. Eqs. (8), (13), did not explain ΔLs.
(4) In rows 162-163, the SMF was cut and then spliced with a piece of HCF, as shown in Fig. 2 (a-b). It should be Fig. 3 (a-b).
(5) Lines 237-240 and 315-316, where the measurement envelopes FSRe for S-1, S-2 and S-3 agreed with the theoretically calculated values, are too absolute and are not exactly the same, and it is suggested that they be revised to be essentially the same with minimal error. Secondly, the text stated that the proposed sensor was fully capable of compensating for temperature variations, here again the statement is too absolute and it is suggested that the conclusion be revised.
(6) Lines 246-247, the envelope inclination produced a red shift in the spectrum, corresponding to an increase in axial strain. It is suggested to explain what the red shift phenomenon is.
(7) In lines 251-252, the measured strain values agreed very well with the theoretical analysis and small errors caused by measurement errors in the equipment can be seen. It is suggested that the strain values from the theoretical analysis be listed and compared with the measured strain values so that the degree of agreement can be better seen. Also, it is recommended to indicate where the small errors caused by the measurement errors of the equipment can be seen.
(8) Figures 9, 10 and 11, R2 of the linear correlation function suggest retaining the same number of decimal places.
(9) Lines 280-281, where it was mentioned in the text that the strain sensitivity of the sensor could be further improved by increasing the bandwidth of the signal source, suggest that this be explained.
(10) Lines 298-300, the text stated that the proposed sensor showed higher sensitivity and a wider strain detection range, however there were sensing structures in Table 2 that were more sensitive than those in the text and some that had a wider test range than those in the text, please suggest a comparative standard to show that the approach in this paper is more convincing.
Author Response
Sep 27, 2022
Sensors mdpi
RESPONSE TO REVIEWER’S COMMENTS
Reviewer 1:
In this paper, the authors presented a highly sensitive strain sensor based on tunable cascaded Fabry-Perot interferometers (FPI), the novelty of which lies in the tuning of the required air reflector cavity for the reference FPI to obtain higher sensitivity and improved performance. Cascaded FPIs comprise a sensing FPI and a reference FPI that effectively produce a Vernier Effect (VE). Unlike conventional interferometry, which monitors the spectrum of a sensing interferometer, VE-based sensors rely on tracking changes in the superimposed spectral envelope, and in optical sensing, VE-based fiber optic sensors are notable for their highly sensitive response to different physical quantities. This paper is somewhat novel and provides a specific study of the subject, but there are still some problems with this paper. The following are the issues in this manuscript.
Q-1: Lines 20-22 and 328-329, the text mentioned that the proposed sensor exhibits an optimum strain sensitivity of 23.9 pm/με over the range 0 με to 3000 με , which is 13.73 times higher than the strain sensitivity of 1.74 pm/με for a single sensing FPI. As can be seen below, by using VE, the S-1 strain sensitivity is 23.9 pm/με , but the strain range for S-1 is 0 με to 2600 με . Not only this, but S-2 and S-3 are not 13.73 times more sensitive than the strain sensitivity of a single sensing FPI. It is recommended that it should be made clear that it is S-1 (375μm) that exhibits the best sensitivity of 23.9 pm/ με in its range, and not all air reflector samples achieve the best strain sensitivity of 23.9 pm/ με .
Answer: We are very thankful for the Reviewer’s constructive feedback, and we have updated the above question in the manuscript as follows.
Experimental results demonstrate that the proposed sensor exhibits optimum strain sensitivity of 23.9 pm/με, 17.54 pm/με, and 14.11 pm/με cascaded with the reference FPI of 375 μm, 365 μm, and 355 μm in cavity length which is 13.73, 10.08, and 8.10 times higher than the single sensing FPI with a strain sensitivity of 1.74 pm/με, respectively.
The Vernier Effect enhances the sensor’s sensitivity 13.73 times that of a single sensing FPI when the reference FPI length is 375 μm.
Q-2: Lines 99-100 and 168, the strain sensitivity range is suggested to be modified to 0 ~ 23.9 pm/ με. It was also mentioned in the text that the strain sensitivity of the proposed sensor is 24 times higher than that of the FBG, and it is suggested that the value of the FBG sensitivity be given. the filament power of the SMF and HCF after splicing is suggested to be modified to 0 ~ 70 W.
Answer: We are very thankful for the Reviewer’s constructive feedback, and we have updated the above question in the manuscript as follows.
The strain sensitivity of the proposed sensor is obtained as 0 ∼23.9 pm/με within the measurement range of 0 and 3000 με, which is approximately 24 times higher than FBG (~1.01 pm/με) [24].
The SMF and HCF are spliced with filament power of 0 ~ 70W, a pre-gap distance of 8.0 μm, a pre-push distance of 5.0 μm, a hot push distance of 14.0 μm, and a filament burning duration of 5 sec.
Q-3: Eqs. (2), (3), did not explain the parameter λ; Eqs. (9), (10), did not explain the parameters λr and λs; Eq. (11), the denominator was not fully displayed. Eqs. (8), (13), did not explain ΔLs.
Answer: We are very thankful for the Reviewer’s constructive feedback, and we have updated the above question in the manuscript as follows.
, (eq2) (2)
, (eq3)
where λ is the operating wavelength.
(eq9)
(eq10)
where λr and λs are the resonant wavelength dips of reference and sensing interferograms, respectively.
(eq11)
(eq8)
where ΔLs is the change in sensing FPI cavity length due to applied strain.
Q-4: In rows 162-163, the SMF was cut and then spliced with a piece of HCF, as shown in Fig. 2 (a-b). It should be Fig. 3 (a-b).
Answer: We are very thankful for the Reviewer’s corrections. We have made the correction as suggested. The update in the manuscript is as follows:
The SMF is cleaved and then spliced with a piece of HCF, as shown in Fig. 3(a-b).
Q-5 Lines 237-240 and 315-316, where the measurement envelopes FSRe for S-1, S-2 and S-3 agreed with the theoretically calculated values, are too absolute and are not exactly the same, and it is suggested that they be revised to be essentially the same with minimal error. Secondly, the text stated that the proposed sensor was fully capable of compensating for temperature variations, here again the statement is too absolute and it is suggested that the conclusion be revised.
Answer: We are very thankful for the Reviewer’s suggestions, as desired the changes in the manuscript are given below:
The measured envelope FSRe of S-1, S-2, and S-3 are 41.45 nm, 31.33 nm, and 24.96 nm as shown in Fig. 8 (a-c), which are closely equivalent to theoretically calculated values as, 43.9 nm, 33.04 nm, and 27.08 nm, respectively. The minimal error is caused by the measuring equipment which is negligible.
The proposed sensor exhibits temperature-strain crosstalk as low as 0.0205 με/°C. It shows that the proposed sensor is fully capable of compensating for temperature variations when it is under different strains (i.e., 0 µε to 3000 µε).
Q-6: Lines 246-247, the envelope inclination produced a red shift in the spectrum, corresponding to an increase in axial strain. It is suggested to explain what the red shift phenomenon is.
Answer: We are very thankful for the Reviewer’s suggestions, as desired the red-shift phenomenon is defined in the manuscript as follows:
The envelope dip produces a red-shift (the change in wavelength toward a longer wavelength) in the spectra corresponding to increasing axial strain.
Q-7 In lines 251-252, the measured strain values agreed very well with the theoretical analysis and small errors caused by measurement errors in the equipment can be seen. It is suggested that the strain values from the theoretical analysis be listed and compared with the measured strain values so that the degree of agreement can be better seen. Also, it is recommended to indicate where the small errors caused by the measurement errors of the equipment can be seen.
Answer: We are very thankful for the Reviewer’s suggestions, as desired we have updated Table 1 displaying the theoretical and experimental values of FSRe and M as follows:
The measured strain values match well with theoretical analysis, and a small error is seen which is induced by equipment measurement error, as given in Table 1.
Table 1. FSR and sensitivity of the proposed sensor
Sample |
FSRr |
FSRs |
Theoretical |
Experimental |
Sensitivity |
|||
FSRe |
Mag. factor M |
FSRe |
Mag. factor M |
|||||
S-1 |
3.20 |
2.985 |
43.90 |
13.71 |
41.45 |
13.73 |
23.90 |
|
S-2 |
3.28 |
2.985 |
33.04 |
10.07 |
31.33 |
10.08 |
17.54 |
|
S-3 |
3.35 |
2.985 |
27.08 |
8.07 |
24.96 |
8.10 |
14.11 |
|
Q-8: Figures 9, 10 and 11, R2 of the linear correlation function suggest retaining the same number of decimal places.
Answer: We are very thankful for the Reviewer’s suggestions. As desired, the values of R2 are changed in Fig. 9, 10, and 11, and make them consistent.
Q-9: Lines 280-281, where it was mentioned in the text that the strain sensitivity of the sensor could be further improved by increasing the bandwidth of the signal source, suggest that this be explained.
Answer: We are very thankful for the Reviewer’s suggestions, answer to the above question is given below:
If we increase the bandwidth, let say 1400 nm to 1600 nm, then by using FSRs = 2.985 nm, FSRr = 3.04 nm (having cavity length 395µm), FSRe = 163 nm, thus it produces 53.7 times higher sensitivity and requires wider wavelength span to demonstrate the different axial strain (i.e. 0 to 3000 µε). So we have used a narrow bandwidth source to investigate the sensor’s performance (For example: when the cavity length of reference FPI is 375um, it only went upto 2600 µε to measure the axial strain), and the following statement given in the manuscript endorses the capability of the sensor considering to the limitation of the wider broadband source.
However, the strain sensitivity of the sensor can be further enhanced by increasing the bandwidth of the source. Nevertheless, the proposed sensor showed a highly sensitive response to strain and provided measurements over a wider range of axial strain by employing a narrow bandwidth of source which is an advantageous feature of the sensor, and the wider range of strain measurement confirms that it is a mechanical robust sensing structure.
Q-10: Lines 298-300, the text stated that the proposed sensor showed higher sensitivity and a wider strain detection range, however there were sensing structures in Table 2 that were more sensitive than those in the text and some that had a wider test range than those in the text, please suggest a comparative standard to show that the approach in this paper is more convincing.
Answer: We are very thankful for the Reviewer’s suggestions. Although, In table 2, ref [19] and ref [23] demonstrate higher strain sensitivity i.e., 37.3 pm/ µε and 47.14 pm/ µε, respectively, but their measurement range is very limited (i.e. 0 to 444 [19] and 0 to 60 [23]) as compared to our proposed sensor. Our sensor confirms its robustness while demonstrating a wider measuring range of axial strain, i.e., 0 to 3000 µε.
The revision in the manuscript is as follows:
A comparison of the proposed sensor is made with earlier reported sensors which are listed in Table 2. It can be ascertained from the comparison that an effective use of VE in sensing architecture can significantly enhance the sensitivity of the sensor, and the proposed sensor demonstrated higher sensitivity with a wider strain detection range. Although, ref. [19,23] reported higher strain sensitivities but presented a limited detection range, i.e., less than 444 µε and 60 µε, respectively. Moreover, the proposed structure is simple and easy to fabricate compared to the reported structures, which required laser inscription [5,7] and difficult fabrication techniques [16,18] for realization. On top of that, the proposed sensor exhibit far superior sensitivity than the reported sensors in ref. [6,8,10,11,14,18] .
We tried our best to improve the manuscript and made suggested changes in the manuscript. These changes will not influence the framework of the paper. We included all the changes and highlighted them in red in the revised paper.
We appreciate Editors/Reviewers’ warm work earnestly and hope that the correction will meet with approval.
Once again, thank you very much for your comments and suggestions.
Sincerely yours,
Dr. Farhan Mumtaz
Department of Electrical and Computer Engineering,
236 Emerson Electric Co. Hall,
Missouri S&T, 301 W. 16th St. Rolla, MO 65409-0040,
mailto: [email protected]
Cell: +1 573 647 3681

Reviewer 2 Report
In this paper, the authors present a highly sensitive strain sensor based on tunable cascaded Fabry-Perot interferometers, Cascaded FPIs consist of a sensing FPI and a reference FPI which effectively generate the Vernier Effect. The sensing FPI comprises a hollow core fiber (HCF) segment sandwiched between single-mode fibers, and the reference FPI consists of a tunable air reflector, which is constituted by a computer programable fiber holding block to adjust the desired cavity length. The results have shown cost-effective, easy to manufacture, repeatable, and shows a highly linear and stable response in the wider range of axial strain. This article is clear, concise, and suitable for the scope of the journal. Several small suggestions are supplied:
1. Suggest the authors improve the introduction part, now the first paragraph has no refs, which is strange.
2. Suggest the authors supply some sentences about the comparison of the proposed sensor with earlier reported sensors.
3. Suggest the authors supply more detail about the HCF used here.
4. The sensor present here may also be useful for marine structural health monitoring, suggest the authors enhance the introduction part with one lastest review on this topic:
Optical fiber sensing for marine environment and marine structural health monitoring: A review Optics and Laser Technology, 2021.
Author Response
Sep 27, 2022
Sensors mdpi
RESPONSE TO REVIEWER’S COMMENTS
Reviewer 2
In this paper, the authors present a highly sensitive strain sensor based on tunable cascaded FabryPerot interferometers, Cascaded FPIs consist of a sensing FPI and a reference FPI which
effectively generate the Vernier Effect. The sensing FPI comprises a hollow core fiber (HCF)
segment sandwiched between single-mode fibers, and the reference FPI consists of a tunable air
reflector, which is constituted by a computer programable fiber holding block to adjust the desired
cavity length. The results have shown cost-effective, easy to manufacture, repeatable, and shows
a highly linear and stable response in the wider range of axial strain. This article is clear, concise,
and suitable for the scope of the journal. Several small suggestions are supplied:
Q-1: Suggest the authors improve the introduction part, now the first paragraph has no refs, which
is strange.
Answer: We are very thankful for the Reviewer’s suggestions, answer to the above question the
references are updated in the first paragraph of the introduction section:
A fiber-optic strain sensor is an optical device that uses optical fiber technology to determine
the strain on an object [1]. The strain experienced by the optical fiber is measured by measuring
changes in the properties of light, such as intensity, wavelength, and the state of polarization. These
miniaturized sensors are based on optical fiber, which is often close to the diameter of human hair
[2]. Due to their small size, fiber-optic sensors find practical applications in compact structures
and tight spaces. Fiber-optic sensors have many advantages over electrical and electronic sensors,
such as compact size, light weight, immunity to electromagnetic interference, low cost, stability,
and high sensitivity [3]. Due to their versatile advantages, they have been made a part of many
practical applications for various industrial sectors such as defense, steel industry, structural health
monitoring, civil engineering, bio-medical, aerospace, and marine engineering [4].
Q-2: Suggest the authors supply some sentences about the comparison of the proposed sensor with
earlier reported sensors.
Answer: We are very thankful for the Reviewer’s suggestions. As recommended the comparison
para is updated in the manuscript as follows:
A comparison of the proposed sensor is made with earlier reported sensors which are listed in
Table 2. It can be ascertained from the comparison that an effective use of VE in sensing
architecture can significantly enhance the sensitivity of the sensor, and the proposed sensor
demonstrated higher sensitivity with a wider strain detection range. Although, ref. [19,23] reported
higher strain sensitivities but presented a limited detection range, i.e., less than 444 µε and 60 µε,
respectively. Moreover, the proposed structure is simple and easy to fabricate compared to the
reported structures, which required laser inscription [5,7] and difficult fabrication techniques
[16,18] for realization. On top of that, the proposed sensor exhibit far superior sensitivity than the
reported sensors in ref. [6,8,10,11,14,18] .
3. Suggest the authors supply more detail about the HCF used here.
Answer: We are very thankful for the Reviewer’s suggestions. The main description and
parameters of HCF are given in section 3 “Fabrication of the sensor”. Which are highlighted
below. However, its microscopic transverse cross-section is given in Fig. 4(d). The changes
made in the manuscript related to HCF is as follows:
An HCF, with inner and outer diameters of 55.1 µm and 125 µm, respectively, is used to
fabricate the sensing FPI. The material of HCF is pure silica and its refractive index is 1.444.
Q-4: The sensor present here may also be useful for marine structural health monitoring, suggest
the authors enhance the introduction part with one lastest review on this topic:
Optical fiber sensing for marine environment and marine structural health monitoring: A review
Optics and Laser Technology, 2021.
Answer: We are very thankful for the Reviewer’s suggestions. As proposed, we have updated
the review reference in the first paragraph of the introduction section.
Due to their versatile advantages, they have been made a part of many practical applications for
various industrial sectors such as defense, steel industry, structural health monitoring, civil
engineering, bio-medical, aerospace, and marine engineering [4].
We tried our best to improve the manuscript and made suggested changes in the manuscript. These
changes will not influence the framework of the paper. We included all the changes and highlighted
them in red in the revised paper.
We appreciate Editors/Reviewers’ warm work earnestly and hope that the correction will meet
with approval.
Once again, thank you very much for your comments and suggestions.
Sincerely yours,
Dr. Farhan Mumtaz
Department of Electrical and Computer Engineering,
236 Emerson Electric Co. Hall,
Missouri S&T, 301 W. 16th St. Rolla, MO 65409-0040,
mailto: [email protected]
Cell: +1 573 647 3681

Reviewer 3 Report
The author's report on an extremely sensitive strain sensor where novelty lies in computer programmable tunable air reflector cavity for improved performance. The tunable cascaded Fabry-Perot Interferometers (FPIs) generate the Vernier Effect in the sensing setup. The sensors sensitivity is enhanced ~13.73 times in comparison to single sensing FPI owing to the Vernier Effect. The senor effectively presents extremely low cross talk with temperature. The easy fabrication process, robustness and cost-effectiveness of the sensor makes it ideal for industrial applications.
The authors present an in-depth and informative introduction with references to the relevant recent literature. The article contains important and significant novel results that will be of great interest for the broad scientific readership of the journal. The results are repeatable making it a reliable sensitive stable strain sensor.
For the reasons mentioned above, I suggest this work deserves publication in sensors, provided the following (minor) comments and suggestions are fulfilled:
1. The authors should define abbreviations at the first instance these appear in the manuscript. For example, VE is not defined in the introduction. (Page 2, line 46)
2. The Figure 3 (page 6) caption should be revised. What does (a) and (b) signify? Figure 3a and Figure 3b should be referenced and discussed in detail in the main text of the manuscript.
3. The authors should add the units to the y-axis shown in Figure 8 (page 9).
4. The authors used three different samples of air reflectors (S-1= 375 µm, S-2= 365 µm, S-3= 355µm). How is the strain sensitivity and bandwidth affected if the sensing FPI cavity length is chosen different then 400 µm?
5. What is the correlation between the core and cladding diameter for SMF and HCF?
Author Response
Sep 27, 2022
Sensors mdpi
RESPONSE TO REVIEWER’S COMMENTS
Reviewer 3
The author's report on an extremely sensitive strain sensor where novelty lies in computer
programmable tunable air reflector cavity for improved performance. The tunable cascaded FabryPerot Interferometers (FPIs) generate the Vernier Effect in the sensing setup. The sensors
sensitivity is enhanced ~13.73 times in comparison to single sensing FPI owing to the Vernier
Effect. The senor effectively presents extremely low cross talk with temperature. The easy
fabrication process, robustness and cost-effectiveness of the sensor makes it ideal for industrial
applications.
The authors present an in-depth and informative introduction with references to the relevant recent
literature. The article contains important and significant novel results that will be of great interest
for the broad scientific readership of the journal. The results are repeatable making it a reliable
sensitive stable strain sensor.
For the reasons mentioned above, I suggest this work deserves publication in sensors, provided
the following (minor) comments and suggestions are fulfilled:
Q-1: The authors should define abbreviations at the first instance these appear in the manuscript.
For example, VE is not defined in the introduction. (Page 2, line 46)
Answer: We are very thankful for the Reviewer’s correction. The vernier effect (VE) abbreviation
is defined in the Abstract line 15
Q-2: The Figure 3 (page 6) caption should be revised. What does (a) and (b) signify? Figure 3a
and Figure 3b should be referenced and discussed in detail in the main text of the manuscript.
Answer: We are very thankful for the Reviewer’s suggestions. We have updated the description
of Fig. 3 as follows:
Figure 3. The fabrication process of Sensing FPI (a) cleaved fibers (b) spliced fibers.
Q-3: The authors should add the units to the y-axis shown in Figure 8 (page 9).
Answer: We are very thankful for the Reviewer’s suggestions. As suggest the Fig. 8 is updated.
Q-4: The authors used three different samples of air reflectors (S-1= 375 µm, S-2= 365 µm, S-3=
355µm). How is the strain sensitivity and bandwidth affected if the sensing FPI cavity length is
chosen different then 400 µm?
Answer: We are very thankful for the Reviewer’s query. Answer to the above query: If we
increase the bandwidth, let say 1400 nm to 1600 nm, then by using FSRs = 2.985 nm, FSRr = 3.04
nm (having cavity length 395µm), FSRe = 163 nm, thus it produces 53.7 times higher sensitivity
and requires wider wavelength span to demonstrate the different axial strain (i.e. 0 to 3000 µε).
So we have used a narrow bandwidth source to investigate the sensor’s performance (For example:
when the cavity length of reference FPI is 375um, it only went upto 2600 µε to measure the axial
strain), and the following statement given in the manuscript endorses the capability of the sensor
considering to the limitation of the wider broadband source.
However, the strain sensitivity of the sensor can be further enhanced by increasing the bandwidth
of the source. Nevertheless, the proposed sensor showed a highly sensitive response to strain and
provided measurements over a wider range of axial strain by employing a narrow bandwidth of
source which is an advantageous feature of the sensor, and the wider range of strain measurement
confirms that it is a mechanical robust sensing structure.
Q-5: What is the correlation between the core and cladding diameter for SMF and HCF?
Answer: We are very thankful for the Reviewer’s suggestions. The correlation of SMF and HCF
in terms of parameters is updated in the manuscript as given below:
An SMF-28e with core and cladding diameter of 8.2 µm and 125 µm and refractive indices of
1.4682 and 1.4672, respectively, is used as a lead-in fiber. An HCF, with inner and outer diameters
of 55.1 µm and 125 µm, respectively, is used to fabricate the sensing FPI. The material of HCF is
pure silica and its refractive index is 1.444.
We tried our best to improve the manuscript and made suggested changes in the manuscript. These
changes will not influence the framework of the paper. We included all the changes and highlighted
them in red in the revised paper.
We appreciate Editors/Reviewers’ warm work earnestly and hope that the correction will meet
with approval.
Once again, thank you very much for your comments and suggestions.
Sincerely yours,
Dr. Farhan Mumtaz
Department of Electrical and Computer Engineering,
236 Emerson Electric Co. Hall,
Missouri S&T, 301 W. 16th St. Rolla, MO 65409-0040,
mailto: [email protected]
Cell: +1 573 647 3681

Round 2
Reviewer 1 Report
The authors have addressed all my comments, and the paper is now acceptable.
Author Response
Thanks for the reviewer's recommendation for the acceptance of the manuscript.